# Porcine Models of Spinal Cord Injury

**DOI:** 10.3390/biomedicines11082202

**Published:** 2023-08-04

**Authors:** Connor A. Wathen, Yohannes G. Ghenbot, Ali K. Ozturk, D. Kacy Cullen, John C. O’Donnell, Dmitriy Petrov

**Affiliations:** 1Center for Brain Injury & Repair, Department of Neurosurgery, Perelman School of Medicine, University of Pennsylvania, Philadelphia, PA 19104, USA; connor.wathen@pennmedicine.upenn.edu (C.A.W.); yohannes.ghenbot@pennmedicine.upenn.edu (Y.G.G.); ali.ozturk@pennmedicine.upenn.edu (A.K.O.); dkacy@pennmedicine.upenn.edu (D.K.C.); odj@pennmedicine.upenn.edu (J.C.O.); 2Center for Neurotrauma, Neurodegeneration & Restoration, Corporal Michael J. Crescenz VA Medical Center, Philadelphia, PA 19104, USA; 3Department of Bioengineering, School of Engineering and Applied Science, University of Pennsylvania, Philadelphia, PA 19104, USA

**Keywords:** spinal cord injury, porcine models, regeneration, translational neurotrauma, hemisection, contusion, balloon compression technique

## Abstract

Large animal models of spinal cord injury may be useful tools in facilitating the development of translational therapies for spinal cord injury (SCI). Porcine models of SCI are of particular interest due to significant anatomic and physiologic similarities to humans. The similar size and functional organization of the porcine spinal cord, for instance, may facilitate more accurate evaluation of axonal regeneration across long distances that more closely resemble the realities of clinical SCI. Furthermore, the porcine cardiovascular system closely resembles that of humans, including at the level of the spinal cord vascular supply. These anatomic and physiologic similarities to humans not only enable more representative SCI models with the ability to accurately evaluate the translational potential of novel therapies, especially biologics, they also facilitate the collection of physiologic data to assess response to therapy in a setting similar to those used in the clinical management of SCI. This review summarizes the current landscape of porcine spinal cord injury research, including the available models, outcome measures, and the strengths, limitations, and alternatives to porcine models. As the number of investigational SCI therapies grow, porcine SCI models provide an attractive platform for the evaluation of promising treatments prior to clinical translation.

## 1. Introduction

Traumatic spinal cord injury (SCI) is a devastating event that leads to varying loss of sensorimotor, autonomic, sphincter, and sexual dysfunction. SCI is a major driver of disability worldwide, afflicting over 20 million individuals [1]. Motor vehicle collisions (MVC), falls, and violence are the most common causes of SCI [2], which disproportionately affect young individuals aged 15 to 29. Injury in this age group accounts for the majority of disability adjusted life years in the SCI population [3]. In recent years, however, there has been an increasing incidence of SCI amongst older adults, potentially driven by global increases in life expectancy [4]. On average, the annual cost of injury is 14.5 billion dollars, which positively correlates with the severity of injury [2].

Spinal cord topography changes along its rostrocaudal and ventrodorsal axes, making the location of injury a major determinant of SCI subtype and severity [5]. For example, high cervical spine injuries may lead to tetraplegia, autonomic dysfunction, and loss of respiratory drive requiring lifelong mechanical ventilation, while thoracic cord injury spares respiratory and upper extremity function [6,7]. Both cervical and thoracic spine injuries are suprasacral injuries and may result in neurogenic detrusor overactivity that has disabling social consequences and increases the risk of urinary tract infections [8]. Given the disability that results from these devastating injuries, substantial efforts have been made to mitigate secondary injury following SCI [9].

Secondary injury stems from a variety of sources including host response to mechanically damaged tissue and pathologic states of spinal cord blood flow that lead to impaired tissue metabolism [10]. Direct cellular injury that occurs from the primary insult also triggers a cascade of damaging events including glutamate-mediated excitotoxicity and oxidative stress [11]. These effects combine to create a hostile environment for recovery, which is made worse by the limited regenerative capacity of the central nervous system. Advances in first responder, hospital, and rehabilitation care have led to dramatic improvements in SCI outcomes [12]. However, meaningful recovery of the sensorimotor and autonomic function that injured patients desire have remained elusive and are an active area of preclinical investigation [9,10,11].

Animal models have been developed using different species and mechanisms of injury to characterize pathogenesis of SCI and test new therapies. Unfortunately, many promising preclinical treatments have failed to show efficacy in clinical trials [13]. For example, pharmacologic therapies such as high-dose methylprednisolone and GM-1 ganglioside are not recommended by current guidelines since randomized clinical trials have failed to show sufficient benefit [14]. One potential reason for translational failure is the inadequacy of widely used rodent models of SCI, which account for 72% of SCI models [13]. Porcine models of SCI account for 1.5% of animal SCI studies and are a promising alternative species that may hold translational relevance due to comparable spinal anatomy and cardiovascular physiology. In this article, we review the current state of porcine models of SCI.

## 2. SCI Pathophysiology

The majority of SCI results from sudden, traumatic impact on the spine causing fractured or dislocated vertebrae. The impacts on the spinal cord can be broken down into the primary injury, which occurs due to direct mechanical forces at the time of injury, and the secondary injury, which is the result of downstream effects including ischemic injury, excitotoxicity, and inflammatory damage [11].

The primary injury can be classified based on mechanism of trauma, including (1) impact plus persistent compression, (2) impact alone with only transient compression, (3) distraction, and (4) laceration/transection [15]. Impact leads to destabilization of the spinal column as a result of bony and ligamentous injuries. Bone and ligamentous injuries can cause further direct injury to the spinal cord through fragments compressing the cord or ligamentous instability causing shearing or stretching injuries. Distraction injuries occur when two adjacent vertebrae are pulled apart, and laceration and transection injuries can occur through missile injuries, severe dislocations, or sharp bone fragment dislocations. Intraparenchymal, subdural, or epidural hematomas may also form and further compound injury from bony compression [11].

The primary injury then sets off a cascade of pathophysiologic mechanisms leading to further morbidity and mortality in SCI patients. Numerous secondary injury processes are initiated following the primary insult. In addition to the direct damage to the cord parenchyma, the compressive forces on the spinal cord can compromise spinal cord perfusion leading to tissue ischemia and further cell death [16,17]. Disruption of the brain spinal cord barrier facilitates the formation of cytotoxic and vasogenic edema in addition to an influx of peripheral inflammatory cells. A complex interaction of these processes result in apoptosis, necrosis, axonal degeneration, gliotic and fibrotic scar formation, and demyelination, which contribute to the persistent neurological deficits experienced by patients with SCI [18].

## 3. Broad Overview of SCI Models

Development of new SCI therapeutics is dependent on effective preclinical models that can be used to develop and evaluate new treatments. The first widely publicized SCI model was developed by Alfred Allen at the University of Pennsylvania [19]. In 1911, he published the first report on a canine model of SCI via controlled weight drop on the exposed spinal cord. Since his early experiments, numerous methods have been developed to induce experimental SCI across several species [13]. Experimental methods are broadly categorized as contusion, compression, and transection injuries. Contusion models are the most frequently studied, followed by transection and compression models [13]. Although cervical SCI has the highest clinical prevalence, most SCI models employ thoracic injuries (81%), a practice that is frequently attributed to ethical and resource concerns with cervical SCI-associated morbidity [13]. Rodent models of SCI are the most frequently cited species in the published literature, accounting for 88.4% of studies (72.4% rat, 16% mouse) [13].

Large animal models of SCI are less frequently reported in the literature. The first porcine model of SCI was developed in 1996, 85 years after Allen’s canine SCI report [20]. Prior to development of porcine and nonhuman primate models, small animal models of SCI using rats and mice predominated due to low cost, small size, a well-characterized genome, and established functional assessment tools. However, clear anatomic and physiologic differences between rodents and humans present major limitations when generalizing insights from the rodent SCI literature to the human SCI population. Differences in the functional organization of the rodent spinal cord, vascular supply to the spinal cord, size of the spinal column, and the well-described potential for spontaneous recovery following SCI in rodents limit translatability of these models [21]. In a review of animal SCI models by Shari-Alhoseini et al., only 1.5% of over 2000 SCI studies reported use of porcine models with another 1.5% using nonhuman primate (NHP) models [13]. In comparison, 72.4% of studies used rats, 6% mice, 2.4% rabbit, 2.3% dog, 2.2% cat, and 0.4% goat, sheep, or bovine models [13]. Despite their infrequent use in the published literature, large animal models remain important in the study of SCI and development and preclinical testing of novel treatment strategies.

Comparative studies of spinal cord structure and function across species have found a significant degree of homology between the porcine and human spinal cord and column anatomy when compared to rodents [22,23,24]. A more accurate recapitulation of the human spinal cord is of great translational importance. Ongoing strategies for reanimation of limbs following SCI include implanting tissue engineered axonal constructs to bypass the region of SCI [25], neuromodulation to convert cortical motor intent into action via implantable spinal cord electrodes [26], and fiberoptic monitoring of spinal cord blood flow and oxygenation for closed-loop blood pressure augmentation using epidural probes [27]. Translation of these implantable technologies is limited by scalability when using small animal models. Additionally, the lateral location of the corticospinal tract in pigs is more similar to the corticospinal tract (CST) in humans, whereas rodent CST is divided into dorsal, ventral, and lateral components [24]. An increased level of similarity in the functional organization and size of the relevant anatomic tracts is essential in determining the efficacy of new therapies [28].

## 4. Porcine Models of SCI

Similar to rodent models of SCI, porcine spinal cord injuries are experimentally produced by contusion, compression, selective spinal tractotomy, or transection of the spinal cord. These injuries produce varying degrees of axotomy and axonal regeneration [29]. Neuronal death, vascular damage, and connective tissue scarring are differentially impacted based on mechanism of injury, allowing experimental modeling of the varied presentations of human SCIs. Below, we discuss porcine models of SCI, with summaries provided in Table 1.

### 4.1. Contusion Models

Contusion models of SCI remain the most widely utilized in porcine models. In a recent systematic review, 70% of published studies of porcine SCI used contusion-based injuries, 43% weight drop with subsequent compression, 16% weight drop alone, and 11% using modified computer-controlled impactors similar to controlled cortical impaction in traumatic brain injury [35]. Contusion-induced injuries first require a laminectomy at the level of injury to expose the dura and underlying spinal cord [30]. To secure the device platform to the animal, pedicle screws can be placed to mount the device. After ensuring appropriate positioning of the guide rail and impactor overtop the laminectomy defect, the impactor is dropped from a predetermined height to then impact the cord. After injury, the impactor may be left in place and additional weight may be added to simulate ongoing spinal cord compression after the initial injury. The severity of injury may be adjusted by changing the mass of the impactor, the height from which it is dropped, and the duration of compression after initial impact [36].

To reduce the variability in size and severity of lesions produced by the weight drop method, Zuchner et al. developed a spring-loaded impactor device equipped with a load sensor to more accurately estimate the force of impact applied during a given injury [37]. Although the authors encountered a degree of variability in the injuries produced with this device, efforts to mitigate that variability through optimizing positioning of the animals during surgery, rigidly fixating the spine to minimize dissipation of force through extension of the spine in response to the impact, and standardization of the treated levels via preoperative X-ray substantially reduced this variation [37]. 

Kuluz et al. also described the use of a controlled cortical impactor (CCI) device to induce SCI in piglets [38]. In this study, a 6 mm impactor tip was used to injure the spinal cord of 3–5-week-old piglets in which the average spinal cord diameter was 5.5 to 6.5 mm. Complete and incomplete injuries could be selectively and reliably obtained by varying the depth of impact and pressure generated.

### 4.2. Compression Models

Spinal cord compression injuries account for the next most commonly cited porcine SCI model with balloon compression models being cited in 5% of studies and surgical clip application in 6% [35]. The compression model of SCI was first introduced in the 1950s by Tarlov in a canine model [39]. Tarlov’s device consisted of a hydraulic device that inflated a bulb-shaped balloon with either water or iodinated contrast that was inserted into the spinal canal after a laminectomy was performed at the site of injury. Acute or chronic compression injuries were induced by variation in the time course of balloon inflation. 

Foditsch et al. described a similar method of inducing SCI in minipigs via minimally invasive techniques. A needle is introduced percutaneously into the lumbar epidural space which facilitates placement of a guidewire and serial dilations before insertion of a kyphoplasty balloon that is threaded into the thoracic epidural space before inflation. This minimally invasive, percutaneous method has the advantage of inducing less pain to the animal. The technique also avoids the need for upfront laminectomy in contusion and transection methods, which increases translatability as injury does not occur to a decompressed spinal cord in clinical practice. 

In addition to balloon compression, variations of the rodent clip compression SCI have also been studied in porcine models. In the clip application model, a laminectomy is performed at the level to be injured, followed by the placement of a calibrated clip, such as an aneurysm clip, around the cord. Given the size of the porcine spinal cord, these models have been adapted with the use of specially developed devices capable of providing precisely controlled compressive forces. Injury severity can be modulated by using clips that are calibrated to deliver a different amount of force or by adjusting the length of time the clip is left in place on the spinal cord. Zurita et al. described a technique in which a durotomy is performed prior to application of two Heifetz’s clips directly onto the spinal cord [40]. Following the surgical procedure, the authors reported the formation of a reproducible necrotic centromedullary lesion 2 weeks following injury. Similarly, Kowalski et al. showed reproducible injuries after epidural application of Heifetz’s clips for 30 min [41]. While the clip application method is notable for its ease of implementation and consistent delivery of a calibrated force, the force applied by the clips produces a predominantly laterally directed compression, unlike the typical dorsal or ventral compressive forces generated in clinical SCI.

### 4.3. Transection Models

Unlike rodent models, in which hemisection and complete transection injuries are commonly utilized, such methods are reported at a significantly lower rate in porcine models [36]. The damage caused by transection of the spinal cord in animal models is representative of lacerating spinal cord injury in humans, which may account for up to 20% of SCI [42]. Similar to the weight drop method, a laminectomy is first required to expose the dura and underlying spinal cord in order to perform hemisection or complete transection of the porcine spinal cord. 

Many animal SCI models produce complete paraplegia. However, given the large number of human injuries meeting criteria for incomplete SCI (American Spinal Injury Association Grades B–E), hemisection of the spinal cord offers an injury mechanism that can mimic the deficits associated with incomplete SCI [13]. Prior work in other animal models has demonstrated that hemisection of the cord causes a reliable pattern of neural damage with significant neuronal loss in the spinal cord tissue adjacent to the lesion and chronic motor disability [43]. Hemisection of the spinal cord can produce injury phenotypes similar to Brown–Séquard syndrome in human patients, but also hold the potential to induce monoparesis of the upper extremity, which maybe a useful tool for studying cervical lesions while minimizing the morbidity associated with more severe, tetraplegia-inducing complete cervical injuries [32]. Injury phenotypes for hemisection of the cervical spinal cord included severe and chronic paresis of the ipsilateral forelimb with substantial recovery of the hindlimb, consistent with the motor syndrome observed in humans with asymmetric SCI [32]. 

Another spinal cord transection model that has been described is a transection caudal to the last sacral spinal cord segment [44]. In this model, a spinal cord injury was induced in the sacrocaudal spinal cord of Yucatan minipigs to cause paralysis of the tail while sparing pelvic limb, rectal, and bladder function. Dorsal laminectomy of the seventh lumbar and first two sacral vertebrae is performed, and the spinal cord is then transected at the junction of the last sacral and first caudal spinal cord segment using tenotomy scissors. This spinal cord transection model has been utilized for cellular transplantation research and offers a novel method for investigating the effect of cellular transplantation on axonal regeneration and functional recovery. Transection of the sacral spinal cord provides the benefit of producing paralysis of the tail only, allowing for a more humane model than those leading to severe thoracolumbar SCIs in pigs. In addition, this reduces the practical nursing challenges associated with inducing pelvic limb and bladder paralysis in a large mammal. However, as motor impairment and bowel and bladder dysfunction are the most frequently cited drivers of disability in patients, the sacrocaudal model may prove inadequate in evaluating the efficacy of potential therapeutics targeted towards those functional domains.

In a study designed to investigate the early postinjury response of sympathetic nerve activity following high cervical injuries, Ruggiero et al. described a complete transection at the C1 level in a non-survival injury model [45]. Given the significant anesthetic and physiologic support required to maintain the animals following this type of injury, the widespread applicability of this model remains limited to studies of the hyper-acute period after injury, which may ultimately limit frequent utilization of this technique. Overall, as large animal studies are utilized as a more advanced step on the path to translation, the significant incongruencies between clinical SCI pathophysiology and that induced by cord transection models may explain why few studies have been described in the porcine SCI literature.

### 4.4. Ischemic Models

Spinal cord injury from ischemia may occur following fractures that compress the anterior spinal artery causing anterior cord syndrome, aortic surgery, and thromboembolic phenomena during endovascular procedures. Comparable vascular anatomy between humans and pigs have made pigs an ideal candidate to study spinal cord ischemia in the area of aortic surgery research [35].

Recently, a porcine model of pure spinal cord ischemia (i.e., no compression) was used to test SCBF monitoring devices [27]. Busch et al. performed a lumbar laminectomy with placement of a spinal cord blood flow (SCBF) monitoring probe in the epidural space that utilized diffuse correlation spectroscopy (DCS) to measure changes in SCBF. This noninvasive optical technique measures fluctuations in near infrared light (NIR) due to red blood cell (RBC) motion. Ischemic injury was produced by placing a REBOA balloon catheter into the aorta via femoral artery cannulation. Sensors on the probe were found to have a sensitivity of 0.87 and specificity of 0.91 for detecting a 25% decrement in SCBF below the level of aortic occlusion [27]. Monitoring of SCBF is also important in contusion models of SCI as spinal cord ischemia is a driver of secondary injury. For example, prior porcine models of SCI that have used laser doppler flowmetry (LDF) to measure SCBF following moderate trauma from weight drop measured a 47.5–61.1% reduction in SCBF following injury [46]. 

Interestingly, when Busch et al. compared DCS to LDF in the porcine model of spinal cord ischemia, reperfusion after releasing aortic occlusion measured using DCS transiently increased above baseline values, whereas LDF returned to baseline. This may reflect differences in sampling, as LDF can only measure surface microcirculation, while DCS penetrates deeper microcirculation [27]. The ability to monitor deeper tissues may be especially important in incomplete spinal cord injuries such as central cord syndrome where damage primarily occurs in the central grey matter with white matter sparing, resulting in segmental sensorimotor dysfunction without long tract injury. 

Thus, ischemic models of SCI can be used to prevent ischemia during vascular procedures, investigate ideal pressors to optimize spinal cord blood flow [47], and serve as a platform for testing devices that allow targeted hyperdynamic therapy following traumatic SCI. Indeed, current clinical guidelines following traumatic SCI recommend maintaining mean arterial pressure (MAP) between 85 and 90 mm Hg for 5–7 days following SCI to mitigate secondary injury from hypoperfusion, but little is known on how this strategy affects SCBF across individuals [48].

### 4.5. Other Models

Penetrating SCI accounts for only ~5% of traumatic SCI. However, such injuries disproportionately affect young male patients, which leads to significant disability costs due to lost productivity and larger duration of long-term care needs. SCI resulting from gunshot wounds likely display different pathophysiologic characteristics than blunt injuries as a result of the unique processes of ballistic injury and cavitation. Given their large size, pigs provide a much more suitable model system to evaluate pathophysiologic mechanisms following penetrating SCI [20]. Few studies employing porcine models of penetrating trauma have been published, which reflects the relative rarity of this entity. Given the relative infrequency of penetrating SCI and the significant ethical concerns of such models, their larger utility remains in doubt.

## 5. Porcine Strains Used in SCI Models

There are several wildtype porcine strains used in medical research. While domestic farm pigs were one of the first porcine strains to be used, their large size presents significant challenges in the study of SCI, as adults frequently achieve weights of greater than 200 kg [49,50]. This has led to exploration of minipig strains as more favorable for biomedical research purposes. Minipigs offer significant advantages for biomedical research given their more manageable size and gentle disposition, which simplify the logistics of animal care while providing a more accurate representation of human spine biomechanics [51,52]. The Yucatan porcine strain is most widely used within SCI research, followed by the Gottingen mini, Vietnamese potbellied mini, and Yorkshire pigs [35].

The Yucatan minipig is a hairless and docile pig that reaches a weight of 70 kg (154 lbs) [52]. The Yucatan pig is an inbred strain, leading to less genetic variability [50]. This is important when considering genetic manipulations in SCI research and responses to new drug therapies. A genetically defined population is advantageous in animal research when compared to outbred strains that have more genetic variability [53]. 

In contrast, the Gottingen minipig offers an even smaller alternative, reaching up to 38 kg (83 lbs) in weight [50]. Size differences in porcine strains are important, as researchers must balance replicating human height and weight with research expenditures (e.g., housing size, food, and handling).

As the majority of porcine SCI injury models require laminectomy to expose the cord prior to contusion, transection, or compression, the effect of the laminectomy on subsequent spinal instability must be evaluated and addressed due to potential confounding effect of iatrogenic SCI secondary to mechanical instability. These size considerations make domestic farm pigs most suitable for non-survival or short-term studies, as demonstrated by Ruggerio et al [45].

As with any animal model, the use of porcine strains for biomedical research should be conducted in accordance with legal and ethical principles. In addition, researchers should be careful to follow strain-specific considerations such as acclimation periods and individual housing following surgical procedures [54].

## 6. Strengths of Porcine Models

Due to the similarity of porcine spinal cord morphology and physiology to humans, porcine models offer significant advantages for preclinical therapeutic trials for human disease. Pig spinal cords are similar in size, dimensions, vertebral body height, and circulatory system to humans [22,55,56]. Toossi et al. recently published a comparative study examining the anatomy of the lumbosacral spina cord in humans and domestics pigs, in addition to Sprague–Dawley rats, rhesus macaques, and cats [57]. Relative to the other species studied, there were significantly greater similarities between several metrics between humans and pigs including the length of the lumbosacral enlargement, cross-sectional area of the spinal cord, and morphology of the central gray matter. This varies across porcine strains, as minipigs have a spinal cord ~½ the diameter of humans [58]. As clinical SCI can sometimes span multiple vertebral levels, the increased length of the porcine spinal cord allows for the study of axonal regeneration across large distances more commensurate with humans, an essential factor in determining the translational potential of bioengineered SCI therapies [28]. Additionally, the position of the porcine corticospinal tract is lateral to the central gray matter, similar to humans [24]. Such anatomic considerations make porcine models better suited to study the effects of implantable therapies on motor recovery compared to rodent models. With respect to sensory pathways, nociceptive neurons traverse the ventrolateral spinal cord in pigs [59]. The location of these fibers correspond with the ventrolateral positions of the anterior and lateral spinothalamic tracts in humans, which are the primary tracts responsible for conveying nociceptive input to the thalamus. Especially in porcine models of SCI, in which the larger caliber of the spinal cord can magnify the regional differences in pathology based upon the injury model employed, a more accurate arrangement of critical functional pathways is helpful in establishing the translational relevance of both the injury model itself and any therapies investigated within a given model.

The vascular anatomy of the porcine spinal cord also shares significant similarities to humans [56]. Vascular compression and damage lead to tissue ischemia and subsequent reperfusion injury, which are important components in the acute phase of SCI. In the chronic phase, subsequent angiogenesis occurs and plays a further role in the chronic remodeling that occurs after SCI. As injury patterns may vary significantly depending upon differences in vascular supply to the cord as well as the mechanism of injury employed in a given study, this homology is an important factor in the interpretation of the effects of tissue ischemia following experimental SCI. In addition to the similarities in spinal cord vascular anatomy, the circulatory system of the pig also more closely approximates the human than many other large animal models. The well-characterized cardiovascular system of the pig also makes it an attractive model in the study of SCI-induced spinal shock in the acute phase of SCI, and long-term cardiovascular dysfunction that results in the chronic phase of SCI [60].

Recovery of voluntary bladder control is cited as a high priority for spinal cord-injured patients, and pigs have comparable lower urinary tract anatomy and physiology. Anatomic studies in pigs have shown comparable features including slit-like urethral openings and similar urethral epithelial lining, although differences in smooth and striated detrusor muscle arrangement exist [61]. Importantly, humans and pigs share similar voiding activity as both species have relaxed bladders during filling stages prior to detrusor contraction. Awake urodynamic studies in Yucatan minipigs undergoing contusion–compression injury were recently demonstrated by Keung et al., which demonstrated similar pathophysiologic changes in detrusor activity following spinal cord injury [62]. Compared to healthy cohorts, spinal cord-injured pigs displayed detrusor contractions during filling stages, which mimics neurogenic detrusor overactivity seen in spinal cord-injured humans [62]. Importantly, pigs did not display recovery of spontaneous voiding weeks after injury that can be seen in rodent models.

Several studies have investigated the neurophysiologic changes following both complete and incomplete SCI in pigs [38,63,64,65]. Numerous outcome measures have been described in pigs including histologic; electrophysiologic; CSF sampling; microdialysis within spinal cord parenchyma; spinal cord pressure and perfusion; multiple imaging modalities including MRI, CT, and ultrasound; as well as behavioral outcomes, especially motor function scoring [35]. The wide array of available neurophysiologic outcomes and significant similarities between human and porcine SCI have thus enabled the rigorous study of both clinically accepted and investigational therapies in attempts to better understand the mechanisms through which these treatments may affect recovery after SCI. For example, Zurita et al. showed that functional locomotor recovery in pigs after transplantation of bone marrow stromal cells (BMSC) into an experimental SCI lesion was paralleled by the recovery of somatosensory-evoked potentials, reduction in lesion size upon MRI, and by the formation of tissue-containing axon bundles mixed with differentiated BMSCs expressing a variety of markers including those of both neuronal and glial lineages [63]. Hu et al. described the different neurophysiologic profiles of pigs who sustained complete versus incomplete SCI induced by balloon compression. In their study, motor-evoked potentials (MEPs), SSEPs, and novel “spine-to-spine-evoked spinal cord potentials” (SP-EPs), which were generated by direct stimulation of the spinal cord and subsequent measuring of potentials at more distal segments located both above and below the injured level [65]. They found that in pigs that received complete SCI lesions, MEPs, SSEPs, and SP-EPs were all completely diminished without recovery. In animals who received incomplete lesions, however, while MEPs and SP-Eps were significantly impaired, SSEPs remained unchanged throughout the operative procedure. Additionally, in the incomplete group, SP-EPs recovered partially in some subjects, although not back to their preinjury baseline. These findings are consistent with findings in human electrophysiologic research in which SSEPs may be preserved even in patients with significant neurologic deficits, while MEPs and D-waves (directly recorded spinal cord impulses, similar to the SP-EPs described by HU et al.) are much more sensitive and specific predictors of subsequent neurologic deficits [66].

In addition, pigs are the only large omnivore in which complex transgenic manipulations have been successfully performed [67]. Transgenic manipulation of porcine lines opens the door for a wide variety of experimental designs that require reliable reporter gene expression, immunocompromised phenotypes, or manipulation of other genes of interest. 

## 7. Limitations of Porcine Models

One important limitation of ungulate models, such as porcine SCI models, is the inability to study upper extremity deficits and recovery, a major driver of disability and frequently cited desire for recovery by SCI patients [68]. Consideration must be made for more suitable models, particularly nonhuman primates, for studies aiming to evaluate upper extremity recovery with a high degree of translational potential.

As a result of the less frequent utilization of porcine SCI models, there is less standardization in quantification of behavioral outcomes [35]. While the Porcine Thoracic Injury Behavior Scale (PTIBS) is commonly used [30], numerous others are frequently cited, including the Porcine Neurological Motor (PNM) Score [31], Miami Porcine Walking Scale [69], Tarlov Scale [63], Individual Limb Motor Scale [32], and Quadruped Position Global Scale [32]. Comparisons across studies of porcine models of SCI are thus impaired by the variety of scoring systems described in the literature. Of the frequently used scoring systems, the PTIBS offers advantages over other available scoring systems. The 10-point scale allows for detailed hindlimb function scoring while also grouping scores into gross locomotor function—hindlimb dragging (score 1–3), stepping (score 4–6), impaired walking (score 7–9), and normal function (score 10) [30]. Additionally, the PTIBS has good inter- and intraobserver reliability.

The detailed 14-point PNM grading system evaluates movement of the tail and movements across the bilateral hip, knee, and ankle joints in the hind limbs [31]. Prior reports using this scale have not documented inter- and intrarater reliability and have required experienced scorers to complete the task. This limits generalizability of the PNM scale [31,70]. The original 5-point Tarlov scale was created in the 1980s to evaluate hindlimb function in rats following SCI and has been used across species with modifications into a short 4-point and longer 6- and 10-point scales [40,71]. Modifications to the scale make it difficult to perform direct comparisons between SCI studies.

In addition to the variation in motor outcome scales reported in the literature, evaluation of sensory function and pain are highly limited in porcine models relative to other animals [72]. Chronic neuropathic pain is an important factor negatively influencing quality of life after SCI and a thorough understanding of how any proposed therapies will modulate that pain is critical in assessing the translational potential of any new treatment. Assessment of pain in swine is limited and past porcine models of SCI have used vocalization in response to pressure applied to the back as a crude measure of allodynia. To maximize the data obtained from porcine studies of SCI, development and standardization evaluations across multiple functional domains is essential.

## 8. Other Large Animal Models

NHP represent the primary alternative to swine for large animal SCI investigations. One systematic review of tissue engineering approaches to SCI showed NHP models were used in 3.2% of identified studies [28]. Another, larger systematic review, evaluating all published animal models of SCI, not just those limited to tissue engineered therapies, similarly found a low frequency of NHP studies, with only 1.5% of over 2000 publications reporting their use. In comparison, 77.4% used rat models and 1.9% used porcine models. NHP models of SCI larger mirror the techniques used in swine and rats. These include contusion via weight drop or mechanical impaction, various transection methods, and compression with balloons or clips.

The benefits of NHP models include decreased genetic interspecies differences relative to humans [73], larger size and more similar neuroanatomic organization of the spinal cord relative to rodents [74], and the potential to evaluate more advanced motor behaviors including bipedal locomotion and hand dexterity [75].

However, NHPs are subject to similar barriers to adoption as other large animal models [76]. Similar to pigs, NHPs are more expensive and require specialized housing and veterinary care. Furthermore, NHPs are subject to a greater degree of ethical concerns given their higher degree of phylogenetic similarity to humans. To our knowledge, while these considerations do not preclude their use in SCI research, it is important to ensure that the selection of such models is done only after the careful exclusion of other available animal models.

Historically, canines served as one of the first SCI model systems as reported by Allen’s seminal work detailing his device for facilitating contusive SCI via weight drop [19]. Tarlov also developed the balloon compression model in canines [39]. More recently, Fukuda et al. refined this by detailing methods for a laminectomy-free balloon compression injury in mixed breed dogs by insertion of a balloon catheter through the intervertebral foramen [77]. As with pigs, the larger size of canines provide a more direct comparison to human SCI and facilitates more detailed histologic examinations. In addition, canines are more docile, which facilitates examination of neurologic function. Consequently, canine models have been cited more frequently in the SCI literature relative to porcine models: 2.2% vs. 1.5% of published studies, respectively.

## 9. Conclusions

While large animal models have been used since Allen published the first report of experimental SCI in canines, they have largely been supplanted by rodent models. However, a renewed interest in large animal models has been growing as a potential tool to further validate new SCI therapeutics in light of prior translational failures. Perhaps the only translational success in animal research following spinal cord injury is hyperdynamic therapy to improve spinal cord perfusion, as reduced spinal cord blow flow has been demonstrated response to injury across species [78,79].

Porcine models of SCI are of particular interest due to significant anatomic and physiologic similarities to humans. The similar size and functional organization of the porcine spinal cord, for instance, may facilitate more accurate evaluation of axonal regeneration across long distances that more closely resemble the realities of clinical SCI. Furthermore, the porcine cardiovascular system closely resembles that of humans, including at the level of the spinal cord vascular supply. These anatomic and physiologic similarities to humans not only enable more representative SCI models with the ability to accurately evaluate the translational potential of novel therapies—especially biomaterial and cell-based therapies—they also facilitate the collection of physiologic data to assess response to therapy in a setting similar to those used in the clinical management of SCI. The collection of such data can significantly aid in both translation of novel therapies, but also may provide insights into optimizing clinical treatment strategies. Furthermore, relative to NHP models of SCI, the primary large animal model alternative used to test promising small molecule and immune therapies targeting secondary injury, porcine models are less expensive and have significantly less ethical objections [8].

However, the benefits of large animal models with respect to improving translational success remain theoretical. No empirical data exist to support the claim that therapies that prove efficacious in swine are more likely to show benefit in clinical trials relative to therapies evaluated in rodent models alone. While extrapolations may be made from data in other fields where clinical trial outcomes demonstrated a significantly higher degree of concordance with data derived from porcine studies relative to rodent studies, the highly complex and heterogeneous pathophysiologic mechanisms that occur after SCI may present a larger hurdle to overcome. Given this uncertainty, careful consideration must be taken in determining the role of these models in the development pipeline for SCI therapeutics. The situations in which it is appropriate to employ porcine models of SCI must also be clarified. Due to their increased associated costs, reserving these models for the evaluation of therapies that have shown efficacy in lower-order animal models is prudent. While there may be specific circumstances unique to a given study that may drive the selection of one strain, method of injury, or outcome measure over another, efforts to standardize these measures should be undertaken to facilitate cross-study comparisons. Such efforts have the potential to significantly increase the value and utility of investigations utilizing porcine models. 

## Figures and Tables

**Table 1 biomedicines-11-02202-t001:** Porcine spinal cord injury models—strengths and limitations.

Porcine Spinal Cord Injury Models—Strengths and Limitations			
	Strengths	Limitations	Clinical Translation
Contusion only [30]	Titratable force applied can produce graded injury that has been shown to correlate with clinical and histopathologic changes. Cavitation, axonal loss, and syrinx formation mimic sequela of human SCI.	Requires laminectomy, decompressing the spinal cord before injury. Lateral cord displacement during impaction may produce variable injury patterns. Invasive, laminectomy +/− instrumentation requires large surgical exposures and more postoperative care.	Model for central cord where spinal cord is contused during extension from disc–osteophyte complex or compressed from tricompartmental stenosis.
Compression [31]	Force and duration of compression are titratable that can create graded changes in clinical exam. Cavitation, axonal loss, and syrinx formation mimic sequela of human SCI. Procedure can be performed with minimally invasive technique using balloon compression.	Compression without contusion has little clinical translation in traumatic spinal cord injury.	Model for pathology that slowly develops such as degenerative stenosis.
Contusion–compression [30]	Best mimics real-life traumatic SCI. Titratable force applied can produce graded injury that has been shown to correlate with clinical and histopathologic changes.	Mimics initial force against cord and time under compression prior to surgical intervention. Lateral cord displacement during impaction may produce variable injury patterns. Invasive, laminectomy +/− instrumentation requires large surgical exposures and more post operative care.	Model for fracture dislocation injury and central cord with ongoing degenerative stenosis.
Transection [32]	Highly selective injury. Produces clinical and histologic changes similar to hemisection of the human spinal cord.	Rare pathology. Non-titratable injury.	Stab wounds.
Ischemic [27,33,34]	Duration and degree of ischemia can be titrated to grade both clinical injury and neuronal cell death on histology. Ischemia can be induced in a minimally invasive fashion through aortic balloon occlusion. Devices used to measure SCBF can be used in traumatic SCI applications to characterize and treat pathologic changes in SCBF to reduce secondary injury.	Prolonged ischemic time may damage nonneural tissue such as the limbs and kidneys, producing widespread damage and increasing animal care needs.	Iatrogenic injury during aortic surgery. Anterior spinal artery compression from fracture fragments. Blood pressure augmentation following spinal cord injury.
Penetrating [20]	Useful for the study of pathophysiologic mechanisms of injury that can provide insights for penetrating injuries in humans, which disproportionately affect young healthy individuals.	Ethical issues. Difficult reproducibility.	Military and civilian SCI from direct and blast-related injury.

## Data Availability

Not applicable.

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
