# Peer review of "Porcine Models of Spinal Cord Injury"

_biomedicines, 2023, doi:10.3390/biomedicines11082202_

Round 1

Reviewer 1 Report

Wathen and colleagues wrote a review on porcine models of spinal cord injury (SCI). The topic of SCI is important as it causes significant negative impact to individuals, families and society. As larger animal models are important compared to rodents as mention in the review, therefore, the review is of interest to readers. However, a major concern is that the review does not differ or may even be less informative than other published reviews on porcine SCI models, such as Kim et al., 2018 (PMID: 30196652); Weber-Levine et al., 2022 (http://doi.org/10.1089/neur.2022.0038); Ahmed et al., 2023 (https://doi.org/10.1016/j.expneurol.2022.114267). Therefore, this review does not provide novel insight/opinion or information to the porcine SCI model.

As this review predominantly focus on the anatomical and physiological benefits of porcine with human, potential improvement may be to discuss about the pathophysiology in SCI. As there are severeal different version of porcine strains, what are the pros and cons of using them. As there are other large animals as mentioned in line 91, how does the porcine compared to the other large animal species? As authors mentioned the various behavioural scoring systems, they could discuss more on this and provide their opinion to which is better/worst and why. Also, expand more on discussing the study of pain and whether the porcine SCI model is good.

Line 242 - 249, comparison of blood flow techniques can be done in non-porcine SCI models, so not specific for porcine SCI models. 

No major concerns.

Reviewer 2 Report

Dear Authors. 

The work submitted for review is very interesting and presents in a systematic way the methodology for the study of core damage using animal models. However, in order for this work to be useful to other researchers it should be improved:

1. add a tabular listing of various spinal cord injury models with a description of their disadvantages, advantages and purpose.

2. make a graphic showing the different methods, with the damaged or destroyed sternae marked on the spinal cord cross sections for each method.

Reviewer 3 Report

The authors of this Review comprehensively, convincingly, and almost editorially flawlessly persuaded about the legitimacy and usefulness of researching the experimental model of porcine spinal cord injury.

Minor suggestions can be provided:

1.      Keyword: delete "swine", correct „balloon compression technique”

2.      I understand, that the Review refers to animal models, but the Authors mention translational medicine, so please provide some examples of refs. when studies on animals influenced the development of neuroscience knowledge in patients after incomplete SCI. 

3.      Moreover, may it be interesting to mention neurophysiological studies on animals including the porcine iSCI model, if they exist, whose results provide significant insight into the mechanism of iSCI functional regeneration. Busscher et al. and Schomberg et al. relevant papers on the porcine anatomy (especially related to the structures of the nervous system) are mentioned, are there any more relevant available?

4.      Put refs in square brackets before dots throughout the text. Avoid different types of referring e.g. in lines 55,58, check throughout the text.

5.      Pros of the experimental model of porcine spinal cord injury are listed, cons are not mentioned.

6.      Using experimental animals for the researchers they have to be conducted in accordance with the Declaration of Helsinki. Ethical considerations have to also be in agreement with Directive 2010/63/EU of the European Parliament and of the Council of September 22, 2010, on the protection of animals used for scientific purposes. These are important, especially during experiments on rats. Are there any special legal regulations regarding the utilization of the porcine models? Legal aspects are abandoned in this Review.

7.      Busscher et al. and Schomberg et al. relevant papers on the porcine anatomy (especially related to the structures of the nervous system) are mentioned, are there any more latest available?

References are selected accurately for the topic undertaken in this Review.

Round 2

Reviewer 1 Report

The authors have addressed all my concerns, so no further comment.

Reviewer 2 Report

Dear Authors. 

Thank you for including a table comparing different models of spinal cord injury. In my opinion, thanks to this the paper is more readable.

I analyzed the cited papers again. Most of them have good quality graphics that make the work easier to read.  Please improve the formatting of the literature in the paper.